# Recent Development of Functional Chitosan-Based Hydrogels for Pharmaceutical and Biomedical Applications

**DOI:** 10.3390/gels9040277

**Published:** 2023-03-27

**Authors:** Siriporn Taokaew, Worasak Kaewkong, Worawut Kriangkrai

**Affiliations:** 1Department of Materials Science and Bioengineering, School of Engineering, Nagaoka University of Technology, Nagaoka 940-2188, Japan; 2Department of Biochemistry, Faculty of Medical Science, Naresuan University, Phitsanulok 65000, Thailand; worasakk@nu.ac.th; 3Department of Pharmaceutical Technology, Faculty of Pharmaceutical Sciences, Naresuan University, Phitsanulok 65000, Thailand

**Keywords:** chitosan, multifunctional hydrogels, fabrication, biomedical applications, controlled drug delivery

## Abstract

Chitosan is a promising naturally derived polysaccharide to be used in hydrogel forms for pharmaceutical and biomedical applications. The multifunctional chitosan-based hydrogels have attractive properties such as the ability to encapsulate, carry, and release the drug, biocompatibility, biodegradability, and non-immunogenicity. In this review, the advanced functions of the chitosan-based hydrogels are summarized, with emphasis on fabrications and resultant properties reported in literature from the recent decade. The recent progress in the applications of drug delivery, tissue engineering, disease treatments, and biosensors are reviewed. Current challenges and future development direction of the chitosan-based hydrogels for pharmaceutical and biomedical applications are prospected.

## 1. Introduction

Recent advances in biomedical engineering afford new regenerative medicine strategies for the most beneficial medical treatments. The localized drug delivery, rapid tissue regeneration, and effective disease treatment are the principal goals of the regenerative medicine. This can be achieved through the precise control over delivery behaviors of cells and/or personalized medicine to match the individual patient. For the delivery carriers, hydrogels serve as a potential candidate and have aroused a great research interest [1,2].

Hydrogels are three-dimensional polymeric networks that can hold a large amount of water or biological fluids, with an extensive applicability in biomedical purposes, including the use as stimuli-responsive sensor materials [3]. The development of hydrogel designs involves supramolecular chemistry, via a crosslinking reaction, to generate new smart hydrogels that are able to respond to environmental stimuli or artificial triggers for drug encapsulation/release, as well as the capability to control the supramolecular morphology and biological activity [4,5]. Compared to hydrogels prepared from synthetic sources, biopolymeric hydrogels are more advantageous due to their biocompatibility, in terms of similarities with extracellular matrix (ECM), and their good biological performance [6]. Among the biopolymeric hydrogels, chitosan-based hydrogels are a class of crosslinked materials intensely studied for their pharmaceutical and biomedical applications over the past decade.

Chitosan is a linear polysaccharide composed of randomly distributed β-(1-4)-linked D-glucosamine and N-acetyl-D-glucosamine units. It is produced by the deacetylation reaction of chitin, which is found in the skeletal structure of crustaceans, insects, mushrooms, and the cell wall of fungi [7]. Chitosan is a versatile natural polymer containing plenty of amino groups, making it hydrophilic, cationic, and reactive, facilitating functional modification, biomolecule encapsulation, muco-adhesiveness [8,9], and adjuvanticity [10]. Moreover, chitosan has anti-cancer effects by inhibiting the upregulated programmed cell death-ligand 1 (PD-L1) expression induced by interferon γ in various tumors via the adenosine 5′-monophosphate-activated protein kinase (AMPK) activation [11]. In order to stabilize the network for some time and to modulate the properties suitable for a selected application, the chitosan hydrogels can be crosslinked physically or covalently, depending on the cross-linking agent [12,13]. However, for a new class of robust hydrogels, a combination of both mechanisms is majorly reported in the current studies.

Over the past decade, functionalization of chitosan and crosslinking into hydrogels with superb self-healing behavior, high elasticity, and significant response properties has become a challenging and fascinating topic. Several pieces of literature have proposed a series of methods to fabricate chitosan-based hydrogels and have proven high applicability in pharmaceutics and biomedical engineering. Therefore, this review summarized recent advances in the preparation of chitosan-based hydrogels with a focus on the updated applications of drug delivery, tissue engineering, disease treatments, and biosensors.

## 2. Gelation Techniques of Chitosan-Based Hydrogels

### 2.1. Synthesis of Water-Soluble Chitosan Derivatives for Self-Forming Gel 

Chemical modification plays a key role in resolving the disadvantage of chitosan, i.e., low solubility in water at neutral pH. The typical chitosan is only dissolved in diluted acid aqueous solution due to the deprotonation of amine group at pH above 6.5, so this restricts the diversified application, especially in the biomedical field. In particular, low dissolution in cell culture medium at physiological pH also limits its cell-related applications. However, the reactivity of one amine group and two hydroxyl groups at C2, C3, and C6, respectively, on anhydroglucosamine units of chitosan polymer chain enables a broad range of reactions employed for functionalization [13]. To endow chitosan with solubility in neutral pH solution and to facilitate the hydrogel formation, various derivatives have been synthesized by chemically grafting with ferulic acid [9], phosphorylcholine [14], 4-imidazolecarboxaldehyde [15], and succinic anhydride [16,17,18]. On the other hand, the chitosan hydrogels can be prepared by the introduction of acrylate groups using 2-hydroxyethyl methacrylate [19,20]. Tailoring the chemical modification of chitosan hydrogels to achieve properties such as drug release involves substitution-degree optimization, functional group pattern, water–polymer ratio, and drug load [21].

Quaternization is one of the methods used to improve the solubility and charge density of chitosan by substituting hydroxyl and amino groups in the glucosamine unit. For the preparation of quaternized chitosan, the typical chitosan is grafted with N-2-hydroxypropyl trimethylammonium chloride [22], 3-Chloro-2-hydroxypropyltrimethylammonium chloride [23], glycidyl trimethylammonium chloride [24], glycidyl triethylammonium chloride [25,26], and 2,3-epoxypropyl trimethylammonium chloride [27]. This modified chitosan is used as a base material for self-healing wound dressing [24,28], such as quaternized chitosan/tannic acid/oxidant hyaluronic acid hydrogel (Figure 1A).

Carboxyethyl chitosan (CEC) is also used for the development of hydrogels with self-healing properties to provide the advantages of minimally invasive surgery and prolong their lifespan. In drug and cell deliveries, CEC solution is mixed with solutions of benzaldehyde groups, capped poly(ethylene glycol) [29], and dextran-graft-aniline tetramer-graft-4-formylbenzoic acid [30], thereby leading to the formation of a dynamic Schiff base bond under physiological conditions (neutral pH at 37 °C). The self-healing happens after 3 h (Figure 1B). Moreover, for self-healing properties, orotic acid-modified chitosan is prepared by condensation reactions between the carboxyl group on orotic acid and the amino group on chitosan. The CO-NH-CO group of pyrimidine skeleton on orotic acid then interacts with 2,6-Diaminopurine (DAP) to form a stable hydrogen-bonded structure. The self-healing happens after 30 min and completes after 1.5 h (Figure 1C). Besides, carboxymethyl chitosan (CMC) is prepared by introducing the hydrophilic group, -CH_2_COOH, on hydroxyl sites of the glucosamine units of the chitosan molecular chains, similar to CEC. Recent research has shown that the hydrogel based-CMC is non-toxic and maintains a moist wound environment because of the hydrophilic carboxymethyl group introduction [31]. Besides the desirable water-solubility of CMC, the advantageous biological properties, such as cell growth promotion and good hemostatic effects, have been reported [32,33]. To improve the applicability in the biomedical field, the recent research has also studied the mechanical property improvement technique by crosslinking with calcium chloride [34,35]. However, the pure CMC film is highly swollen and unstable in the calcium chloride solution, indicating a weak interaction between CMC and Ca^2+^ [36]. Therefore, other CMC crosslinkers are used, such as sodium alginate [36,37,38], oxidized sodium alginate [39], and genipin [40]. In the case of CMC/alginate, increasing crosslinking improves the degradation rate because of a larger number of hydrophilic NH_2_ and COOH groups in the hydrogel matrix [41]. Among these, dextran (oxidized dextran) is an effective crosslinker used for CMC hydrogel preparation due to abundance of adjacent hydroxyl groups, which can be oxidized to generate aldehyde groups and form Schiff base bonds with amino groups of dextran [42,43].

**Figure 1 gels-09-00277-f001:**
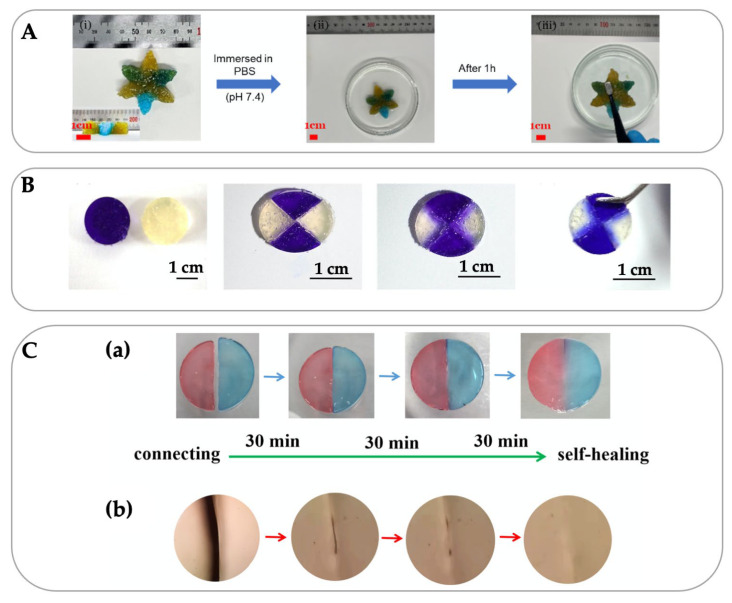
Self-healing property of quaternized chitosan/tannic acid/oxidant hyaluronic acid hydrogel after 1 h (**A**) [reprinted with permission from ref. [24] Copyright © 2022, Elsevier], carboxyethyl chitosan/dibenzaldehyde-terminated poly(ethylene glycol) hydrogel after 3 h (**B**) [reprinted with permission from ref. [29] Copyright © 2017, Elsevier], and orotic acid-modified chitosan/2,6-diaminopurine hydrogel after 1.5 h (**C**) under macroscopic states (**a**) and microscopic states with a microscope (**b**) [reprinted with permission from ref. [44] Copyright © 2022, Elsevier].

Guanidinylation reaction of chitosan is a chemical modification of chitosan using cyanoguanidine as a modifying agent. The introduction of biguanidinium groups onto the backbone of chitosan allows the improvement of solubility, as well as thermal and antimicrobial properties, due to the increase in ionic and hydrogen bonding interactions. The high density of positive charges of biguanidinium groups on the chitosan derivative can interact with the cell membranes of microorganisms, leading to their death. The enhanced antimicrobial property of CMC caused by the guanidinylation reaction is also reported [45]. Moreover, the biguanide-derived chitosan can accumulate in the mitochondria to decrease the high dosage of a biguanide drug to sensitize oxygen-sensitive tumor therapies through oxidative phosphorylation (OXPHOS) inhibition [46].

Other water-soluble chitosan derivatives that have been studied as candidates in biomedical applications, due to their thermogelation, are hexanoyl glycol chitosan synthesized by n-hexanoylation of glycol chitosan [47,48,49], hydroxybutyl chitosan synthesized by conjugation of hydroxybutyl groups to the hydroxyl, and amino groups of chitosan. These derivatives are able to form a thermosensitive hydrogel at over 25 °C by self-crosslinking without additional cross-linking agents, such as initiators or light sources [50,51,52]. Moreover, properties of chitosan and a synthetic polymer can be combined by the graft radical polymerization method to produce hybrid copolymers and apply them in a specific application [53]. Apryatina et al. [54] prepared the copolymers of chitosan with 2-hydroxyethyl methacrylate (vinyl monomers) and N-vinylpyrrolidone by the graft polymerization method, and they studied the conformational transition of chitosan macromolecules (coil-rigid rod and rigid rod-globule) in aqueous acidic solution by spectrophotometry. The coil-rigid rod conformation transition depends on the pH of its surroundings, but it does not relate to the molecular weight of chitosan. This conformation transition is sensitive to temperature. In a solution with pH near 4, the chitosan macromolecules are more flexible, so they appear in the coil conformation. At pH around 5.57, the amino groups of chitosan, which are weakly protonated, form intermolecular bonds with the oxygen of adjacent chains. Therefore, the macromolecules are rigid at this pH [55]. The conformation of chitosan affects the modification, as well as the properties, of the final products. Their study found that yield of the graft copolymer-based chitosan was higher in the case of the coil conformation (96%) than in the rigid rod conformation (81%). It may be relevant to the effective diameter since the rigid rod conformation (34–39 nm) is larger than that of the coil conformation (22–25 nm) [54]. Similarly, Mochalova et. al. [56] also found that the polymerization of 2-hydroxyethyl methacrylate and styrene in chitosan solutions was influenced by the pH of the reaction medium. When the pH of the reaction mixture changed from 4.2 to 5.5, the degree of polymerization increased from 13.4 to 27% w/w.

With hydrophilic chemical group enhancement and/or low molecular weight, these chitosan derivatives have better water solubility, but after the gel is self-forming, an unstable structure causes them to be rapidly broken. Therefore, other materials/crosslinking agents must be added in gelation to improve the physicochemical and biological characteristics of the hydrogels. With high molecular weight, the molecules in the chitosan matrix are more closely crosslinked, thereby enhancing stability of the network structure [57].

### 2.2. Physical Crosslinking

Chitosan-based physical hydrogels are produced by ion-complex formations with negatively charged molecules such as dicarboxylic acids [58], citric acid (tricarboxylic acid) [59], sulfates, phosphates [60], gallic acid [61], alginate [62,63,64,65], pectin [66,67], sulfoethyl cellulose [68], and hyaluronic acid [69]. Through the physical interaction between anions of these molecules and cations of chitosan hydrogels can be formed. While retaining the properties, such as nontoxicity, pH-responsive, hydrophilicity, and biodegradability, of the individual original polymers, e.g., chitosan and hyaluronic acid, the chemical, physical, and other functional properties of the hydrogel are different [70,71]. Moreover, polyelectrolyte complexes, such as chondroitin sulfate [72], carrageenan [73,74,75], and carboxymethyl cellulose [76,77], or other water-soluble nonionic polymers, such as polyvinyl alcohol (PVA) [78,79,80], can also form polymeric bonds with chitosan, resulting in the hydrogel. By amide bond crosslinking between the carboxylated PVA and chitosan, as well as coordination crosslinking between chitosan and Fe^3+^, mechanical, adhesive, and self-healing properties of the hydrogel can be enhanced [81].

Polyol salts, such as glycerol or glucose-phosphate salts, effectively transform chitosan solutions into temperature-sensitive chitosan solutions by allowing the chitosan solutions to remain liquid at room temperature and turn into the gel at body temperature [82]. The main advantage of this thermosensitive system and obtaining thermosensitive hydrogels is their noninvasive injection into the pathologically affected area, while cells and therapeutic agents can be comprised into the solution before in situ injection. This injectable hydrogel also has a shape that matches to the damaged tissue. Among the salts, tripolyphosphate is the most employed as an ionically cross-linking agent of chitosan, due to the high net negative charges per monomeric unit (ranging from 1–5 depending on pH) to obtain chitosan micro/nano hydrogels [83,84,85,86,87,88]. Lapitsky et al. [89,90] demonstrated that not only tripolyphosphate (sodium tripolyphosphate pentabasic; Na_5_P_3_O_10_) but pyrophosphate (sodium pyrophosphate tetrabasic; Na_4_P_2_O_7_) could also boost the ionotropic gelation of chitosan to obtain stable nanogel suspensions. However, the tripolyphosphate has a higher affinity with chitosan chains compared to the pyrophosphate. This leads to lower mechanical properties of chitosan/pyrophosphate hydrogel (shear modulus of ~1 kPa) with respect to the chitosan/tripolyphosphate hydrogel (shear modulus of ~6 kPa) [91], similar to the use of trisodium phosphate (sodium phosphate tribasic dodecahydrate) as a crosslinker [92]. The tripolyphosphate gels show a higher connectivity in the micronetwork, so it has a limited degradation in simulated physiological media up to 6 weeks, whereas pyrophosphate gels degrade almost immediately [91].

The mixtures of chitosan with β-glycerophosphate disodium or glycerol phosphate disodium (βGP), as well as with uridine 5′-monophosphate disodium, are a thermo-gelling system for cell injection, remaining liquid mixtures at physiological pH and turning into gels at body temperature [93,94]. A model of heat-induced transfer of protons from chitosan to βGP, in solution, was proposed by Filion et al. [95] to explain this sol–gel transition. The electrostatic crosslinking, or dehydration effect, occurs due to the increased hydrophobicity of βGP with heat. The mechanism of chitosan gel formation is that the proton transfer from chitosan to βGP is induced by heating, resulting in the neutralization of chitosan and precipitation into a physical hydrogel. The βGP is free to diffuse out of the gel after the gel formation since a continued presence as a proton acceptor is not required. Grinberg et al. [96] studied phase transitions of thermo-responsive chitosan in aqueous solutions, induced by βGP, by means of high-sensitivity differential scanning calorimetry. They found that βGP solutions of chitosan underwent a first-order phase transition upon heating. Binding βGP to the chitosan matrix induces the formation of a highly ordered hydrate structure of chitosan at low temperatures. The binding complexes are under the action of large entropic stress because of this ordered structure. Upon heating, dehydration and dissociation of the complexes occur. Chitosan chains eventually tend to self-associate and form a gel by losing their charges and hydrated structure [96]. However, increasing this salt concentration to reach neutral pH values increases the osmolarity of the mixture to cytotoxic values. Therefore, other salts, with and without mixing with βGP, are used to provide the physiological osmolarity of 300 ± 30 mOsm/L [97]. The salts that have been studied as gelling agents with a combination of βGP are sodium hydrogen carbonate [98,99,100,101] and ammonium hydrogen phosphate [97,102]

### 2.3. Chemical Crosslinking

The primary pathway of crosslinking chitosan to obtain mechanically stable hydrogels is the acid condensation with aldehydes used to form imine bonds (–N=CH–). This condensation of amino groups of chitosan with carbonyl functions to yield imines, i.e., Schiff bases or azomethines. Dialdehydes, e.g., glyoxal and, especially, glutaraldehyde, are often used, but their toxicity to the human body restricts their use for biomedical applications [103]. Therefore, recent studies have focused on monoaldehydes such as fatty aldehydes [104], nitrosalicylaldehyde [105], salicylaldehyde [106,107], and citral [108,109]. The crosslinking using these aldehydes occurs through a hydrophobic interaction (non-covalent interaction), among hydrophobic side chains, on the hydrophilic chitosan backbone [104,110]. Damiri et. al. [110] reported the synthesis of the N-benzyl chitosan by grafting the chitosan with benzaldehyde. The physical cross-linking was obtained via the hydrophobic interactions between aromatic moieties in the backbone. The transparent hydrogel was formed when the temperature was changed from room temperature to 37 °C. The pH value of imine bond cleavage and the hydrogel disassembly rate of the hydrogel can be turned by the selection of CHO/NH_2_ molar ratio [106].

Bratskaya et al. [111,112] considered other factors, which could contribute to stabilization of chitosan hydrogel by using CEC and CMC with different substitution degrees and salicylaldehyde, in the study. Besides hydrophobic interaction (Figure 2A) formed by imine bonds in Figure 2B1, in chitosan and the chitosan derivatives, they also found that reactions between salicylaldehyde and carboxyalkylchitosans yielded stronger hydrogels due to different types of intra and intermolecular hydrogen bonds preferentially formed depending on the type, and degree, of carboxyalkyl substitution, such as hydrogen bonding via C6-hydroxyl group and NH_2_ group (Figure 2B2), via carboxylic group and NH_2_ group (Figure 2B3), and via two carboxylic groups (Figure 2B4). The CEC-salicylimine contained higher free amino groups, able to form all types of hydrogen bonds, providing mechanically stable hydrogel at a salicylaldehyde:polymer molar ratio of 1:10, while gelation in chitosan solution occurred at a much higher molar ratio of 1:2.5 (Figure 2 B5). Due to and reversibility of the C=N bond, the solubility of the CEC hydrogel increased at pH < 4.3 and pH > 7. Moreover, amino acids could be used as chemical affecters for the dissolution of the hydrogel via transimination reaction. As shown in Figure 2C, the solubility of the hydrogel increased in the presence of 20 g/L lysine at pH 8.

In the development of pH-responsive chitosan hydrogels, interactions of water-soluble chitosan derivatives with other aldehydes, e.g., dibenzaldehyde-terminated poly(ethylene glycol) [29], poly(ethylene glycol) diacrylate [113], vanillin [114], and dextran-graft-aniline tetramer-graft-4-formylbenzoic acid [30], have also been proposed as promising strategies. These pH-responsive hydrogels with dynamic benzoic imine covalent bonds represent one of the systems for the controlled release of bioactive compounds. Aromatic aldehydes coupling to chitosan also allow hydrogel formation via temperature-sensitive hydrophobic interactions. Yang et. al. [44] fabricated orotic acid (Vitamin B13)-modified chitosan hydrogel by chemically crosslinking with 2,6-diaminopurine (DAP) in the presence of 1-Ethyl-3-(3-dimethylaminopropyl) carbodiimide hydrochloride (EDC) and N-hydroxysuccinimide (NHS), which were used as catalysts to form amide bonds. The obtained hydrogel has dual responsiveness to temperature and pH, self-healing properties, thermal reversibility, injectability, and conductive sensing.

Even though the mechanical properties and responsive characteristics in different pH levels or temperatures of the chitosan hydrogels are improved by chemically crosslinking, when the hydrogel dissolves, chemical residue can remain and invade surrounding tissue, causing cytotoxic effects. Attempting to purify the hydrogels is time-consuming and increases the cost of the hydrogels. Among chemical crosslinking agents, genipin, extract of *Gardenia Iasminoides Ellis* fruit, has been used to crosslink chitosan because it is a natural crosslinking agent, it is non-toxic, and it has favorable chemical characteristics, thereby obtaining biostable and biocompatible chitosan hydrogels [115,116,117,118].

Also, porous structure of the soft hydrogels can be tailored by varying genipin and chitosan concentrations [119] (Figure 3A,B), as well as deacetylation degree (DDA) and molecular weight (MW) of the chitosan [120] (Figure 3C). More compact microstructures and improved mechanical properties of the chitosan hydrogel are reported by the use of succinic acid and urea, in addition to chemical crosslinking, in the presence of genipin [121]. Therefore, several studies have reported the utilization of the genipin-crosslinked chitosan for anti-bacterial material [122], as a carrier of drugs including 5-fluorouracil [123], cisplatin [124], and cytarabine [125], as well as for the treatment of diabetes [126] and diabetic wounds [127]. However, the drawback of using genipin as a crosslinker is relatively long gelation time [128].

### 2.4. Photo-Crosslinking

Photo-crosslinking is one of the promising chemical crosslinking methods in the presence of a photo-initiator. Due to rapid cure, low curing temperature, in-line production, and ability to tailor the hydrogel, crosslinking chitosan hydrogel by photopolymerization is widely applied to form 3D hydrogels. The photo-crosslinked hydrogels also have injectability features, because of gel formation, after they are photo-polymerized. Photo-responsive chitosan hydrogel can be prepared through functionalization with light-sensitive components/precursors, such as 2-amino ethyl methacrylate [129], methacryloylglycine [130], glycidyl methacrylate [131,132], and methacrylic anhydride [133,134,135]. Then, the functionalized chitosan solution is mixed with a photo-initiator such as 2-hydroxy-2-methylpropiophenone [134] and lithium phenyl-2,4,6-trimethylbenzoylphosphinate [135]. The methacrylic anhydride, which interacts with the amidogen of chitosan through acylation/methacrylation reaction (Figure 4A), is often used in the recent literature due to its solubility and a certain degree of cytotoxicity. However, disadvantage of the functionalization with photo-curable groups, i.e., methacrylate or acrylate, are time-consuming. The reaction can be sped up by microwave heating (800 W) using a chitosan and methacrylic anhydride molar ratio of 1:1, reaction temperature of 80 °C, launch time of 60 s, and reaction time of 10 min [135]. The hydrogel possesses 3D printability, with a high matching degree to the mold (Figure 4B–D) and injectability with transdermal curing (Figure 4E,F) [135,136].

Several photo-crosslinkable thermo-gelling chitosan derivatives have been reported for the purpose of mechanical properties improvement. However, a high synthetic polymer concentration (15% w/w) for thermo-gelation is required, leading to a lack of biocompatibility [137]. The cell proliferation rate is correlated with the higher concentration of chitosan, the more excessive interference to the cell culture microenvironment, so there have been continuing studies to develop the gelling system with less cytotoxicity but better mechanical properties. Cho et al. [47] prepared non-toxic methacrylated hexanoyl glycol chitosan (HGC-MA) hydrogel (4% w/w HGC-MA aqueous solution) by the hexanoylation and methacrylation of glycol chitosan. The solution could be solidified to form a non-flowing transparent gel when the temperature increased to 37 °C. The further UV light exposure, for 15 min, provides the compressive moduli of 100–150 Pa, depending on the content of the methacryl group. Similarly, hydroxybutyl methacrylated (HBC-MA) chitosan hydrogel, prepared by conjugating the hydroxybutylation and methacrylation of chitosan, possesses both thermosensitive and photo-crosslinkable properties [138,139,140,141]. From a 3% w/v HBC-MA aqueous solution in the presence of a photo-initiator and exposure of visible light (450–550 nm), the hydrogel is formed under physiological condition. The crosslink density of the hydrogel can be controlled by temperature and light, resulting in an optimization of pore size (20–80 μm) and mechanical properties (compressive modulus of 50 kPa after the hybrid photo/thermo crosslinking for 30 s) [138]. The compressive modulus is improved, up to 190 kPa, by reinforcement with a chitin whisker [141]. The in vivo subcutaneous injection into mice allowed the thermogel formation to demonstrate low cytotoxicity of the HBC-MA solution. However, volumetric shrinkage occurs when photo-crosslinked HBC-MA hydrogel is immersed in PBS (pH 7.4) at high temperatures > 30 °C. This may limit its application for cell culture or oral drug delivery [139,140]. Considering the lower crosslinking rate, the higher hydrogel swelling rate, and consequently, the higher the drug release, the crosslinking agents facilitate controlling drug release according to the pH of the medium [28]. The pH-responsive swelling and shrinkage prevention at a neutral pH are studied using N-succinyl hydroxybutyl methacrylated chitosan (NS-HBC-MA) [142]. This photo-crosslinked hydrogel of triple-conjugated chitosan exhibits pH-responsive swelling, which varies with pH, temperature, and degree of succinylation. Moreover, the degree of substitution affects hydrogels in terms of network structure, gelation temperature, microstructure, and degradation in a lysozyme, which is present in human tears, saliva, and mucus [131,142]. Although light-curable hydrogels have been studied, accessibility of the light, i.e., UV penetration, limit its application in the deep skin area, e.g., in deep incompressible hemorrhages, when the hydrogel is applied as hemostatic materials. Moreover, a high-intensity UV exposure causes damage to normal tissue.

### 2.5. Hybrid Crosslinking

Chitosan-based hydrogels can be obtained by various crosslinking approaches with different advantages and disadvantages. Although the physical crosslinking is relatively simple, mild, and avoids the use of organic solvents and high temperatures, the physically crosslinked hydrogels possess weak mechanical properties (compressive modulus of less than 10 Pa) [143]. The recent studies attempt to combine the two crosslinking techniques, resulting in dual-network hydrogels with improved mechanical properties, promising to be a reliable route to high performance materials. The dual crosslinking is summarized in Table 1. For instance, chitosan and salts, e.g., hydrated sodium orthophosphate and βGP as ionic crosslinkers, form a physical gel, but its stability and mechanical strength are weak. Therefore, genipin is added as a co-crosslinker to improve the mechanical properties and chemical stability of the hydrogel. Owing to the combination of these physical and chemical crosslinking approaches to form 3D networks, the chitosan hydrogel, crosslinked by both sodium orthophosphate hydrate and genipin, exhibits a storage modulus value (G′) of 195 Pa, which is higher than genipin/chitosan hydrogel (G′ < 100 Pa) or βGP/chitosan hydrogel (G′ < 10 Pa). This dual crosslinked hydrogel also shows a short gelation time (8 min) [144].

Ailincai et. al. [145] prepared iminoboronate chitosan hydrogels by using 2-formylphenylboronic acid as a chitosan crosslinker. The hydrogel structure facilitates a dual crosslinking—a covalent bond via imine forming and a physical bonding. They found that the ortho position of the boric acid residue stabilized the imine linkage through intra-molecular hydrogen bonds via an iminoboronate motif. The combination of 2-formylphenylboronic acid into the hydrogel also enhanced the antifungal activity against yeast. Antibacterial properties of chitosan hydrogels against *Staphylococcus aureus* is improved depending on the molecular weight of chitosan and the plasma treatment time [157]. Hydrophilicity of the hydrogel surface is improved due to the increases in hydrophilic chemical groups (-COOH and -NH_2_) by the treatment [157]. This plasma treatment is used not only for surface functionalization but also facilitating the crosslinking of chitosan and acrylic acid [146]. Besides the acrylic acid that promoted protonated amine formation, there is an increase in the protonated amines by using N_2_ as a working gas, especially at high chitosan concentrations [146].

For the multiple-network hydrogels, Engkagul et. al. [158] prepared the chitosan hydrogels consisting of hyaluronic acid-triazole linkage, chitosan-copper complex, and the polyion complex of chitosan-hyaluronic acid network. The hydrogel exhibited mechanical strength and non-cytotoxicity against chondrocyte cells. Chen et. al. [155,156] cross-linked carboxymethyl chitosan by oxidizing dextran and poly-γ-glutamic acid (Figure 5A) to form a triple-network of intramolecular amide, intermolecular amide, and dynamic Schiff base bonds (Figure 5B). The gelation time was fast (less than 1 min) but low G’. However, the hydrogel exhibited injectability, antibacterial activities against *Escherichia coli* and *Staphylococcus aureus*, 100% L929 cell viability, in vivo biodegradation, in vivo hemostatic ability, and in vivo wound healing ability.

## 3. Pharmaceutical and Biomedical Applications of Chitosan-Based Hydrogels

### 3.1. Drug Delivery

Drug delivery is the process of administering a therapeutic agent to a patient to achieve a therapeutic effect. The aim of drug delivery is to maximize the positive effects of the drug while minimizing any harmful effects it may have. Hydrogels have gained attention as a drug delivery system due to their ability to control the release of drugs over a desired period of time.

Chitosan hydrogels have been widely studied for their potential as a drug delivery system due to their biocompatibility, biodegradability, and ability to control the release of drugs. It can be designed to release drugs in a variety of ways, such as through diffusion, degradation of the hydrogel matrix, or by using pH-sensitive or temperature-sensitive hydrogels. Additionally, the positive charge of chitosan allows for the preparation of negatively charged drug-loaded hydrogels, which can improve the stability of the drug and prevent premature release. Chitosan hydrogels for drug delivery can be categorized based on the method of drug release. Delivery systems and release modes of various drugs from chitosan-based hydrogels are summarized in Table 2.

#### 3.1.1. Sustained Release Systems

Chitosan hydrogels can be designed to release drugs over an extended period of time, providing sustained delivery of the therapeutic agent [88]. Ailincai et al. [109] encapsulated diclofenac sodium salt into chitosan hydrogel. The ionic interactions between this anionic drug and the polycationic hydrogel matrix increased the intermolecular forces and retarded the drug release. The release rate could be adjusted by varying the molar ratio of chitosan and citral. Genipin-crosslinked porous chitosan fiber has been developed for topical application in wound healing [127]. Clemastine fumarate, a selective histamine H_1_ antagonist, is incorporated into the hydrogel to enable its gradual release into the wound over time. This promotes the proliferation, tube formation, and migration of fibroblasts and endothelial cells, resulting in faster healing time.

Chitosan hydrogels have been developed to deliver insulin to treat diabetes [159,160]. A chitosan-amide-crosslinked matrix is prepared from different fatty acids, N-isopropylacrylamide, and 2-acrylamide-2-methylpropane sulfonic acid [159]. The cross-linked hydrogels containing insulin can form the hydrogel at an acidic pH of 1.2 and a temperature above 32 °C. The sustained insulin release mechanisms fit with the Higuchi and Hixson models. Phan et al. [160] conducted insulin-loaded hydrogel beads by a simple dropping technique. The insulin was encapsulated in layered double hydroxides and coated with chitosan and alginate to create the hydrogel beads. The beads successfully protected insulin in acid and released insulin slowly in the small intestine. Metformin nanoparticles, which are antihyperglycemic agents, are incorporated into a chitosan/PVA polymeric composite using the salt leaching technique [161]. This method creates a porous structure that permits the controlled release of the drug, with the rate of release being determined by the size and distribution of the pores within the composite.

Chitosan hydrogels can be prepared in a form that is suitable for injection, enabling drugs to be delivered, in a minimally invasive manner, to a targeted area of the body. Once injected, the hydrogel transforms from a liquid state to a semi-solid or solid gel through a gelation process, which then facilitates the sustained release of drugs at the injection site [42,65,162].

#### 3.1.2. Targeted Delivery Systems

Chitosan hydrogels can be modified to deliver drugs directly to specific tissues or sites. This targeted drug delivery approach also allows for a sustained release of the drug over an extended period of time, reducing the frequency of administration and improving patient compliance. The properties of the hydrogel, such as swelling behavior, viscosity, and elasticity, can be tailored to control the release rate of the drug, allowing for the precise delivery of the therapeutic agents to the desired site.

**Table 2 gels-09-00277-t002:** Delivery systems and release modes of various drugs from chitosan-based hydrogels.

Drug Delivery Systems	Drug	Mode of Release	Composition of Hydrogel	Ref.
Sustained release systems	Clemastine fumarate	Topical route	Genipin/chitosan	[127]
	Dexamethasone	Injection	CMC/oxidation dextran	[42]
	Diclofenac	Oral/topical route	Citral/chitosan	[109]
	Doxorubicin	Oral route	Chitosan/CMC/formadehyde/succinic anhydride	[163]
	Gabapentin	Injection	Chitosan-g-poly(acrylic acid-co-acrylamide)	[162]
	Insulin	Oral route	Layered double hydroxides beads coated with chitosan/alginate	[160]
	Insulin	Injection	Acrylamide-modified chitosan/tripolyphosphate	[159]
	Metformin	Oral route	Chitosan/PVA	[161]
	Valproic acid	Injection	Chitosan/alginate	[65]
Targeted delivery systems	Betamethasone	Ocular routes	Chitosan/dialdehyde starch	[164]
	Curcumin	Buccal route	Nanocapsules coated with chitosan	[165]
	Famotidine	Oral route	Montmorillonite/chitosan	[8]
	Doxorubicin	Injection	Chitosan/folate	[18]
	Lidocaine	Topical route	CMC/alginate	[41]
pH-sensitive systems	Amoxicillin	Oral/topical route	Gamma-irradiated chitosan/PVA	[166]
	Cefotaxime	Oral route	Chitosan/alginate	[167]
	Cytarabine	Oral route	Chitosan/tamarind/poly (methacrylic acid)	[168]
	Diclofenac	Oral route	Chitosan/Iron oxide	[169]
	Doxorubicin	Injection	Chitosan/Iron oxide	[87]
	Doxorubicin	Injection	CEC/dibenzaldehyde-terminated poly(ethylene glycol)	[29]
	Naproxen	Oral route	Bacterial cellulose/chitosan	[170]
	Phenylalanine	Injection	Oxidized hydroxypropyl cellulose/CMC	[171]
Thermosensitive hydrogel systems	2,6-diaminopurine	Oral route/Injection	Orotic acid-modified chitosan	[44]
	Bupivacaine	Injection	Graphene oxide/chitosan	[172]
	Caffeic acid phenethyl ester	Depot	Acetylated CMC	[173]
	Diethyldithiocarbamate and copper ions	Topical route	Chitosan/βGP	[174]
	Doxorubicin/curcumin	Injection	Thiolated chitosan	[175]
	Etanercept	Injection	Chitosan/βGP/Pluronic F-127	[176]
	Paracetamol	Topical route	Chitosan/βGP/genipin	[177]
Herbal drug delivery systems	Berberine	Oral route	Amino acid-g-chitosan/βGP	[178]
	Cannabidiol	Injection	Chitosan/sodium carboxymethylcellulose	[179]
	Carvacrol	Oral route	CMC/alginate	[151]
	Cortex moutan	Injection	N,N,N-trimethyl chitosan	[69]
	Curcumin	Injection	Chitosan-oligoconjugated linoleic acid	[88]
	*Moringa oleifera* leaf extract	Oral route	Chitosan/alginate	[180]
	*Pueraria lobatae*	Oral route	Chitosan/xanthan gum	[181]
	Quercetin	Injection	Chitosan/halloysite/graphitic-carbon nitride	[182]
	Resveratrol	Oral route	Chitosan/PVA	[183]

To deliver drugs to the stomach, chitosan has been employed as bioadhesive beads to improve drug retention in the stomach [8]. These beads contain amino groups, which give them a positive charge, leading to strong interactions with negatively charged gastric mucus, enabling them to adhere to the mucosal surface of the stomach. Montmorillonite has been used to modulate famotidine release, which is caused by a decrease in chitosan chain expansion. Ortega et al. [165] demonstrated the potential of a chitosan-coated targeted drug delivery system, for buccal drug administration, using curcumin nanocapsules. The mucoadhesive properties and zeta potential of the nanocapsules increased, thereby enhancing the local concentration of the drug. For the control of ocular inflammation, an ophthalmic delivery system using chitosan/dialdehyde starch hydrogel is a suitable option for local administration of betamethasone [164].

Chitosan hydrogels have shown potential as targeted drug delivery systems in cancer treatment. By incorporating anti-cancer drugs, such as 5-fluorouracil [123,184], cisplatin [124], cytarabine [125], doxorubicin [18,87], and methotrexate [185], into the hydrogels, the drugs can be delivered directly to cancer cells. The hydrogels can be formulated into various shapes and sizes, such as nanoparticles or microspheres, to enhance the delivery efficiency and reduce the risk of systemic toxicity. The targeted drug delivery helps to increase the effectiveness of cancer treatment while minimizing the adverse effects associated with conventional chemotherapy. For instance, chitosan conjugated with succinic anhydride and folic acid is developed for the targeted delivery of doxorubicin to osteosarcoma cells [18]. The osteosarcoma cells (MG-63) exhibit a higher cellular uptake of doxorubicin, by expressing folate receptors, compared to lung cancer cells (A549), which do not express the folate receptors.

#### 3.1.3. pH-Sensitive Systems

One of the unique properties of chitosan hydrogels is that they can be designed to respond to pH changes, which makes them particularly useful for drug delivery applications since pH is an important factor. When chitosan hydrogels are exposed to an acidic environment, the chitosan chains become protonated, which causes the hydrogel to swell and release the drug. This swelling behavior can be controlled by adjusting the pH of the hydrogel, allowing for precise control over drug release. Barkhordari et al. [87,169] developed a chitosan/iron oxide nanocomposite hydrogel bead with pH and magnetic-responsive properties. The system was created using chitosan, iron oxide nanoparticles, and sodium tripolyphosphate as a crosslinking agent. The drug release behavior of the hydrogel was strongly influenced by the environmental pH and external magnetic field.

Schiff base hydrogels provide a hydrophilic polysaccharide network that is utilized for pH-sensitive applications. The hydrogels are in situ formed via Schiff’s base reaction between partially oxidized bacterial cellulose and chitosan [170]. They are capable of loading naproxen at a high capacity of above 110 mg/g, and they sustain the release for over 24 h. The release of naproxen in simulated gastric fluid is less than that in intestinal fluid. For Schiff base hydrogel of CEC and oxidized hydroxypropyl cellulose, the drug release selectively targets lower pH environments such as the extracellular milieu around cancer cells [171].

The water-in-oil emulsification method has been used to prepare pH-sensitive nanocarriers [186,187]. The nanobeads composed of chitosan and agarose exhibit pH-sensitive releases of drugs, with different drug releases in tumor-simulated and normal cell-simulated buffers. Tran Vo et al. [166] crosslinked chitosan/PVA hydrogels by using gamma irradiation. The higher swelling ratio of hydrogels was observed under acidic conditions, due to the protonation of amino groups.

#### 3.1.4. Temperature-Sensitive Systems

The incorporation of temperature-sensitive polymers into chitosan hydrogels enable them to exhibit temperature-responsive behavior. When the temperature is raised above or below lower/upper critical solution temperature, the temperature-sensitive polymer undergoes a reversible transition from a swollen state to a collapsed state or vice versa. This causes changes in the physical properties of the hydrogel facilitating the drug release.

Wang et. al. [133] developed chitosan hydrogel by UV crosslinking for NIR light-triggered drug delivery. This thermo-responsive chitosan copolymer synthesized by grafting with poly N-isopropylacrylamide, acetylating with methacryloyl groups, and embedding with photothermal carbon. The poly N-isopropylacrylamide endowed chitosan hydrogel with temperature-triggered volume shrinkage and reversible swelling/de-swelling behavior (∼42% shrinkage in response to elevated temperature induced by NIR). Their work shows that the doxorubicin release rate was accelerated about 40 times higher than that from non-irradiated hydrogels. Qiao et al. [177] described an approach for transdermal delivery of analgesic and antipyretic drugs using a silk fabric coated with a thermosensitive hydrogel consisting of βGP, genipin, and chitosan. The proposed dressing provides a temperature-dependent instant release behavior within the first 2 h of application, and it is particularly beneficial for patients with fever or inflammation.

Injectable chitosan-based hydrogels have had promising applications in the field of drug delivery because they become gelling matrices at the injection site, and they can release drugs in a controlled manner [44,172,173,175]. The hydrogel is used to treat periodontitis by forming a depot, allowing for a sustained release of caffeic acid phenethyl ester in the lesion sites [173]. The thermosensitive hydrogel matrix, made of acetylated CMC, effectively inhibits inflammation and promotes bone tissue repair in periodontitis therapy. Etanercept, a tumor necrosis factor-α inhibitor, is incorporated into injectable-thermosensitive hydrogels prepared from chitosan, Pluronic F127, and βGP for the localized treatment of joints in osteoarthritis. The concentration of βGP in the formulation affects the release of the drug [176]. To target and treat a solid tumor, Li et. al. [175] developed a thiolated chitosan hydrogel with a gelation point of 37 °C. By an addition of the liposome-encapsulated curcumin, the gelation time was extended because the negatively charged surface of liposome electrostatic interacted with the positive charge of chitosan. For the treatment of surgical site infections, chitosan/βGP hydrogel is prepared and, then, incorporated with the liposomal formulation of diethyldithiocarbamate and copper ions [174]. The injectable gel kills 98.7% of methicillin-resistant *Staphylococcus aureus* and prevents 99.9% of biofilm formation of *Staphylococcus epidermidis* over 48 h.

#### 3.1.5. Herbal Drug Delivery Systems

The use of hydrogels in the delivery of traditional medicines has shown great potential in overcoming the challenges posed by traditional drug delivery methods. Chitosan hydrogels have emerged as a promising solution to improve the bioavailability, stability, and safety of traditional medicines. This has led to an increase in the effectiveness of medical treatments and has provided a promising alternative for the delivery of traditional medicines. For the treatment of traumatic spinal cord injury, Zhang et. al. [179] demonstrated the use of in situ gelling hydrogels for the local delivery of cannabidiol. The hydrogel has mechanical stability, similar mechanical properties to the spinal cord, and sustainability of cannabidiol delivery for a prolonged period of 72 h. By enhancing mitochondrial biogenesis and increasing neurogenesis, apoptosis in the spinal cord injury is reduced. By loading Cortex Moutan, the pH stability and morphology of N,N,N-trimethyl chitosan-based hydrogel is modified [69]. However, this pH/temperature-responsive hydrogel possesses high cumulative drug release under mild acidic conditions and with low toxicity to human HaCaT keratinocytes. Solid dispersion of *Pueraria lobatae* is used as an active ingredient in chitosan hydrogel for the oral drug delivery system [181]. The drug release can be varied by the increase in polymer and crosslinker concentrations. To improve the stability of herbal ingredients such as carvacrol (monoterpene phenol) [151], quercetin [182], *Moringa oleifera* leaf extract [180], and resveratrol [183], encapsulation in conjugated chitosan hydrogel is the promising approach. To control the release of berberine hydrochloride, amino acids, i.e., l-lysine (Lys), l-glutamate (Glu), and l-glutamine (Gln), are grafted into chitosan/βGP hydrogel using EDC/NHS as a coupling agent to improve solubility in the pH range of 6.0–7.0 [178].

### 3.2. Tissue Engineering

Chitosan hydrogels have gained increasing attention in the field of tissue engineering due to their unique properties, such as high water content, biocompatibility, and ability to deliver therapeutic agents directly to damaged tissues. Additionally, chitosan is biodegradable into non-toxic products, having no adverse effects on cell viability or differentiation.

Chitosan hydrogel has been used as a scaffold for cells to function [70,98,188,189]. Moreover, it is combined with other materials to mimic the ECM, providing a suitable environment for cells to attach, proliferate, and differentiate. For instance, a scaffold of γ-polyglutamic acid/CMC/bacterial cellulose hydrogel, introduced with magnesium ions, exhibits a positive effect on the repair of osteochondral defects, due to mimicking the ECM of bone tissues [190]. The chitosan-hydroxyapatite scaffolds, based on lyophilized platelet-rich fibrin, demonstrate strong mechanical properties to maintain the stability of the bone repair environment and antibacterial properties. Importantly, growth factor releasing from this scaffold is prolonged [191]. Mechanical properties of the hydrogel (prepared from sodium carboxymethylcellulose and chitosan) are also crucial in treating traumatic spinal cord injury, as they affect the cellular response and tissue regeneration [179]. Coburn et al. [103] found that increasing stiffness of glycol-chitosan hydrogel is associated with a heightened level of the anti-inflammatory interleukin (IL)-10, indicating a shift in macrophage behavior towards resolving inflammation. The anti-inflammatory properties of the hydrogel can be enhanced by the incorporation of specific cell types, e.g., human vocal fold fibroblasts within the hydrogel. Therefore, the optimization of the ratio of biomaterials and the choice of crosslinking agents are important factors that can impact cell function within the scaffold [192].

The chitosan hydrogel can absorb and retain large amounts of water, thereby providing a moist environment that promotes wound healing, such as diabetic wound healing [79,127,152]. Incorporation of therapeutic agents, such as growth factors [193,194,195], antibiotics [27,28], and anti-inflammatory agents [152], into hydrogels additionally enhances the therapeutic effects for diabetic wound healing. Fibroblast growth factor is incorporated in benzaldehyde-terminated 4-arm PEG/CMC hydrogel for diabetic wound treatment [193]. The productions of CD31 and CD34 are upregulated, resulting in increased cellular proliferation, epithelialization, collagen production, hair follicle formation, and neovascularization. However, bacterial infections can impede the healing process of wounds and increase the risk of complications. In the presence of bacteria, the wounds are inflamed, leading to delayed healing, chronic wounds, or even sepsis. Therefore, effective wound healing strategies that prevent bacterial infections are essential. Tobramycin, an antibiotic added to quaternized chitosan/oxidized dextran hydrogel, is studied [28]. The Schiff base crosslinking between the tobramycin and oxidized dextran enables the sustained release property, resulting in controlled wound inflammation levels, promoted collagen deposition, vascular generation, and earlier wound closure. The prevention and treatment of surgical site infections using a chitosan-based lipogel containing diethyldithiocarbamate and copper ions are proposed [174]. The hydrogel exhibits antibiofilm activity against Methicillin-resistant Staphylococcus aureus (MRSA) and Staphylococcus epidermidis, and it reduces the viability of the formed biofilms. Due to hemostatic property, tissue adhesion, wound healing, and infection prevention, the chitosan hydrogel possesses great promise as a tissue adhesive for sutureless wound closure in surgical procedures [32,33,154,155,156,196].

### 3.3. Disease Treatment

Many recent studies propose chitosan hydrogels as an alternative strategy for drug or substance delivery, particularly in cancer treatment. Long chemotherapy is required, of which severe side effects may occur. Localized delivery of anticancer drugs helps to increase the drug concentration at the site of action to enhance bioavailability and releasing behavior. In tumor therapy, several chemo-drugs are loaded in chitosan hydrogel-based carriers, which are expected to provide high drug loading, stability, and hydrophilicity. In addition, these hydrogels exhibit many advantages, i.e., good biocompatibility and easy degradation. Applications of hydrogels combined with chemotherapy, photodynamic therapy, photothermal therapy, sonodynamic therapy, chemodynamic therapy, and synergistic therapy also enable promising cancer treatment strategies [197]. For the first-line drug, doxorubicin is chemically conjugated with acrylate chitosan to obtain sustained-release profiles after injection and to treat solid tumors, especially in breast cancer, which requires long chemotherapy management. In triple negative breast cancer (TNBC), which is insensitive to hormone therapy, the acetylated-chitosan hydrogel system is developed for the kinetically controlled delivery of a synergistic drug pair (doxorubicin and gemcitabine). Such a hydrogel affords a desirable combination index, indicating a stronger synergism. In addition, a notable volume reduction in the TNBC spheroids is achieved, whereas free drugs only reduce growth rate [198]. Moreover, 5-Fluorouracil loaded chitosan hydrogel exhibits good physicochemical properties and in vitro sustained release profiles, resulting in greater inhibition of breast cancer cell viability and in vivo antitumor activity with minor side effects [199]. In addition, gefitinib-loaded cellulose acetate butyrate nanoparticles incorporated into chitosan/βGP hydrogels is established for intratumoral administration in mice bearing breast cancer and displayed a greater cytotoxic effect in comparison with a free drug. The intratumorally injected mice show the antitumor efficacy in vivo [200]. Furthermore, not only the chemotherapeutic drugs but also active pharmaceutical or medicinal substances, such as curcumin and chrysin loaded in the alginate/chitosan hydrogel, are reported for the induction of breast and lung cancer cell apoptosis [201].

For neurodegenerative diseases, there are still no valid therapeutic treatments for motor neuron diseases. Thermosensitive chitosan/βGP/sodium hydrogen carbonate hydrogel is developed for the in vitro 3D model study of motor neuron diseases by supporting the viability and differentiation of the neuroblastoma/spinal cord hybrid cell line [101]. Similarly, chitosan-g-oligo (L,L-lactide) copolymer hydrogel proved that multipotency of embryonic-derived neural stem cells can be sustained and is supported with the differentiation of NSCs along the neuronal lineage [202]. For hormonal therapy of Parkinson’s and Alzheimer’s disease, progesterone shows neuroprotective effects. The progesterone-loaded trimethyl chitosan/sodium alginate/sodium tripolyphosphate increases brain progesterone concentrations in vivo [203]. The oxidized tannic acid-modified gold nano-crosslinker is synthesized and used as a chitosan crosslinker for the preparation of the bioactive self-healing hydrogel for Parkinson’s disease treatment. The neural stem cells grown in the hydrogel display long-term proliferation and differentiation, and inflamed NSCs are rescued. This serves as an effective brain injectable implant to treat Parkinson’s disease [204].

For bone disease and rheumatoid arthritis (RA), chitosan-coated semi-interpenetrating polymer network hydrogels, containing sodium alginate and poly(2-ethyl-2-oxazoline), loaded with 2-isopropyl-5-methylphenol (Thymol), shows no cytotoxicity with mouse mesenchymal stem cells (MSCs). The thymol stimulates osteoblastic differentiation by upregulating RUNX2 and activates the osteogenesis through TGF-β/BMP signaling pathway [195]. For microRNAs delivery, miRNAs also play a crucial role in the osteogenic differentiation of MSCs in bone marrow, including, for example, miR-29b. Naked miRNAs are unstable serums and cannot pass across the cell membrane, so miR-29b loaded in graphene oxide-based nanocomplex, functionalized by polyethyleneglycol and polyethylenimine, is encapsulated into chitosan hydrogel. This system is able to promote osteogenic differentiation, allowing bone regeneration both in vitro and in vivo [205]. For natural product delivery to treat bone disease, calcium carbonate microcapsules containing nanohydroxyapatite particle/chitosan/collagen hydrogel have the prolonged release profiles of quercetin, which has been proven for promoting bone regeneration and delivering natural flavonoids [206]. For RA, deoxyribonuclease I (DNase)/oxidized hyaluronic acid/CMC is injectable, degradable, and biocompatible that also reduces inflammatory cytokine expression for alleviating RA symptoms [207]. Moreover, phenethyl isothiocyanate (PEITC), a bioactive phytochemical, exerts chemopreventive, antioxidant, and anti-inflammatory activities via the Nrf-2 pathway, but its limited water solubility and short half-life are the drawbacks. Therefore, PEITC-loaded chitosan/pluronic hydrogel is developed as a thermosensitive injectable. In the adjuvant-induced arthritis (AIA) rat, PEITC-hydrogel injected into the knee joint reduced paw edema, arthritis score, and bone erosion [208]. Furthermore, chitosan/gelatin/βGP-melanin-methotrexate hydrogel exhibits a rapid release of methotrexate instantaneously, when it is injected directly into the paw joint of collagen-induced arthritis mice, followed by irradiation. Conclusively, treated joints of mice are similar to normal joints under Micro-CT analysis [209].

For gastrointestinal diseases, a gelatin methacrylate/oxidized hyaluronic acid/galactosylated chitosan/Fe(III)@TA@IGF-2 200 hydrogel, loaded with insulin-like growth factor 2 (IGF-2), can regenerate the damaged hepatocytes and improve the survival of the damaged human hepatic stellate LX2 cells. In addition, the HNF-4α and transferrin are upregulated, indicating the restored function of the liver [210]. Furthermore, chitosan-N-acetyl-l-cysteine/sodium alginate/tilapia peptide reduces liver injuries in mice caused by alcoholism [211].

For cardiovascular diseases and hematologic disorders, myocardial infarction (MI) has been considered as a major cause of cardiovascular-related deaths. Myocardial regeneration through stem cell transplantation serves as a promising treatment strategy for MI. For cardiac function recovery, gold nanoparticles incorporated into chitosan/silk fibroin hydrogel, having no toxicity on mesenchymal stem cells or cardiac myoblast H9C2 cells, can stimulate cardiomyocyte fiber restoration and promote regenerative activities in myocardial ischemia cells [212]. For the better therapeutic approaches, basic fibroblast growth factor (bFGF)-loaded pyridyl disulfide-modified CMC/reduced BSA (rBSA) hydrogel has a rapid glutathione-triggered degradation behavior and well-defined NIH 3T3 fibroblast cell proliferation. This loading system also minimizes poor clinical efficacy of bFGF, due to proteolytic degradation, low drug accumulation, and severe drug-induced side effects. In addition, the in vivo echocardiography demonstrated that the left ventricular functions are improved by reducing the fibrotic area in the bFGF-hydrogel treated group [213]. Oncostatin M (OSM), a pleiotropic cytokine belonging to interleukin-6 family and possessing an imperative role in cardiomyocyte dedifferentiation, is also loaded into the poly-(chitosan-co-citric acid-co-N-isopropyl acrylamide) hydrogel. Continuous and localized release of OSM, responsive to a precise pH, accelerates angiogenesis and proliferation of cardiomyocytes, inhibits myocardial fibrosis, and improves cardiac function effectively [214]. Injectable hyaluronic acid-chitosan/βGP hydrogel, loaded with MSCs, are implanted in the mouse MI model. The therapeutic effect of the MSCs is considerably improved by indications of enhanced cardiac function, reduced cardiomyocyte apoptosis, and increased vascularization [215]. For cardiomyopathy, the epicardial implantation of acellular chitosan hydrogels in mice is associated with a reversion of cardiac function loss with maximal effects of the acetylation degree. The extent of fibrosis, the cardiomyocyte length-to-width ratio, the genes involved in fibrosis, and the stress are repressed after implantation. This indicates the potential therapeutic approach to heart failure [216]. For enhancing growth of endothelia cells while suppressing proliferation of smooth muscle cells (SMCs), the hydrogel thin films contain multiple functional groups (-SO_3_Na, -COONa, -OH, and -NH_2_) of a modified sulfonated sodium alginate, and chitosan enhances human umbilical vein endothelial cell growth and inhibits effects on human umbilical artery smooth muscle cell proliferation [217]. For bleeding control, a composite hydrogel, formed by the entrapment of vasoconstrictor-potassium aluminum sulfate and coagulation activator-calcium chloride, enhances the rapid blood clotting by accelerating RBC/platelet aggregation and activating the coagulation cascade. Furthermore, in vivo studies on rat liver and femoral artery hemorrhage achieve hemostasis in a shorter time [218].

### 3.4. Biosensor

Towards the development of biosensors, chitosan supports the immobilization of biological molecules displaying residual primary or secondary amine in its structure, such as antibodies. The available primary amine groups also promote the covalent attachment of enzymes and other proteins via crosslinking reactions with chemical agents, such as genipin [219]. A modified fluorine tin oxide electrode, containing a chitosan/genipin hydrogel, is developed for voltammetric determination of IL-6, a major pro-inflammatory cytokine, as a biomarker of sepsis (tested in a septic mice). This biosensor is also highly selective when tested with three inflammatory cytokines such as IL-12, IL-1β, and TNF-α. Due to accurate and robust IL-6 measurements, it is useful for the early diagnosis of inflammatory diseases [220]. As a pH-sensitive part of the sensor designed for use in microfluidic (on-chip) applications, chitosan/tetraethyl orthosilicate hydrogel is prepared and bonded, by oxygen plasma treatment, to a thin layer of polydimethylsiloxane (PDMS), of which the thickness is a variable parameter. The fabricated sensor responses to pH changes and becomes stable in less than 90 s. The sensitivity of 54 and 94 pixel/pH is shown when using the PDMS thickness of 75 and 28 μm, respectively. The detection limit for both thicknesses of PDMS is at 0.24 and 0.11 pH, respectively [221]. For physical motion detection, chitosan-based multi-networked elastomeric hydrogel is prepared using chitosan, MXene, oxidized sodium alginate, polyacrylamide, Fe(III), and poly(3,4-ethylenedioxythiophene)-poly(styrenesulfonate). The hydrogel exhibits a high mechanical strength (0.91 MPa), toughness (2.99 MJ/m3), stretchability (820%), elasticity (>600%), conductivity (1.31 S/m), and stability upon bending or stretching, with ultra-broad sensitivity up to 11-gauge factor [222]. Besides MXene being used as conductive filler, carbon nanotube (CNT) is also used to fabricate a flexible sensor that can detect human movement for fabricating bio-inspired skin-like wearable devices. Hybrid chemically–physically crosslinked polyacrylamide/chitosan/sodium carboxymethyl cellulose/CNT hydrogel is prepared by simple mixing and UV irradiation. This hydrogel exhibits self-adhesion, high stretchability, 3D printability, good electronic conductivity, and high sensitivity to human motion, such as joint movement, breathing, and drinking water [223].

## 4. Conclusions and Future Perspectives

Chitosan-based hydrogels are a functional material and are of great interest as a potential candidate for pharmaceutical and biomedical applications, due to their biocompatibility and biodegradability. They can be efficiently used for drug delivery systems, tissue engineering scaffolds, disease treatment, and biosensors. However, the development of such hydrogels remains a challenge. Mainly, even though chitosan has biological advantages, it can only be dissolved in a mildly acidic aqueous solution. This shortcoming limits the fabrication of the functional hydrogels. The pure chitosan hydrogels also have poor mechanical property, insensitivity under external stimuli, and macroscopic nonuniformity that are necessary to be improved by crosslinking the neat chitosan or, especially, the chitosan derivatives. Hence, functionalization of the chitosan without deteriorating other advantageous properties is still a broad field of research to achieve. In general, the two possible outcomes are deprotonation of amine groups at neutral pH and tunability of those properties. By changing compositions, dosages, and surrounding conditions to meet a specific application scenario, physicochemical/biological properties and morphologies of hydrogels should be simply regulated. This will expand the scope of the application of chitosan hydrogels.

In comparison with the physical crosslinking, the chemical crosslinking proficiently improves the mechanical property and stability of the hydrogels. This leads to slower degradability and the possibility to control the pore size being recommended for in vivo long-term applications. However, their negative effects on the biocompatibility, due to the cytotoxicity of the crosslinking agents, have been the main aspects to consider because the toxicity seriously hinders the applications in the medical care. Additionally, some of the chemically crosslinked hydrogels are so stable and irreversible that they are insensitive to external stimuli and unable to practice self-healing. Therefore, it is still a great challenge to synthesize chitosan-based hydrogels with stability, rapid strain recoverability, and self-healing ability. Despite secondary crosslinking and multiple crosslinking to reduce the risk of toxicity from crosslinking agents and to enhance tunable properties by using the chitosan derivatives, these methods are limited by the complex fabrications and high costs. Therefore, it is imperative that the chitosan-based hydrogel preparations are simplified to support large scale production, and they should comprise a water-soluble formulation with sustained-release characteristics, which are obligatory for the clinical application. In the clinical translation, biocompatibility, safety, effectiveness, and cost minimization of the chitosan-based hydrogels are crucial parameters and should be assessed through a combination of in vitro and in vivo approaches. In the cell-laden hydrogel systems, the optimization for minimal time to expand, harvest, and process the cells, as well as minimal cell concentration in the hydrogel, are the essential facets to shorten the laboratory-based work and to minimize the cost. Animal models are the gold standard for pre-clinical evaluation of skin sensitization, inflammatory response in subcutaneous implants, and subacute systemic toxicity using rodent models. However, minimization of the cost from preclinical animal models remains an important challenge that need to be overcome.

With appropriate gelation time, mechanical property, swelling, and self-healing, the cell-laden chitosan-based hydrogels will be clinically satisfactory, especially as injectable hydrogels that have been introduced as 3D scaffolds for cell culture and drug carriers. Even though successful cell cultures via the 3D scaffolds for human tissue repair have been found, the prevention of severe hydrogel deformation during cell functions, e.g., migration, proliferation, and differentiation, are necessary. The anti-shrinkage ability and morphological stability of the hydrogel 3D scaffold, therefore, need to be further improved. The development of 3D fabrication methods also facilitates simulating an environment similar to inside the body in order to evaluate drug effectiveness and/or study single-cell behavior, for instance, by microfluidic devices. This microfluidic system needs to be equipped with sensors to monitor and control environmental parameters, such as pH, temperature, fluid flow rate, enzyme, and other parameters involved in cell metabolisms. As this system is complex, network connectivity issues between soft sensors, circuits, and machines needs multi-disciplinary fields such as biological/materials science and electrical/computer engineering to achieve the microfluidic chips integrated with sensors that are high efficiency but low cost and low mass/power consumptions.

## Figures and Tables

**Figure 2 gels-09-00277-f002:**
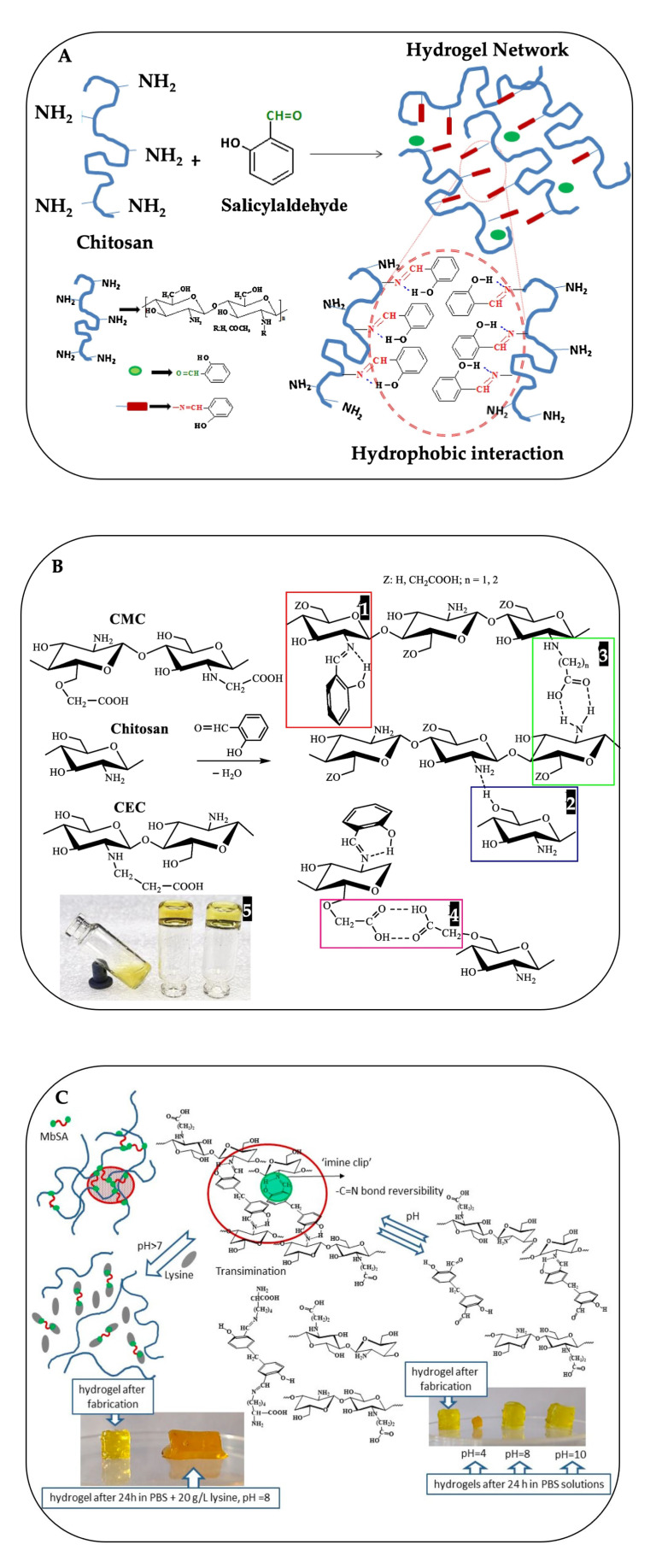
Synthesis of the salicyl-imine-chitosan hydrogels (**A**) [reprinted with permission from ref. [106] Copyright © 2017, Elsevier], chemical structures of chitosan, and its carboxyalkyl derivatives (carboxymethylchitosan; CMC and carboxyethylchitosan; CEC), with a hypothetical scheme of hydrogen bonding in solutions of their salicylimines (**B1**–**B4**) and a photo of gels formed in 72 h after addition of salicylaldehyde to solutions of chitosan, CMC, and CEC (from left to right) at a salicylaldehyde:polymer molar ratio of 1:5 (**B5**) (reprinted with permission from [111] Copyright © 2021, Elsevier) and a scheme of pH-and lysine-induced disassembly of CEC hydrogels cross-linked with methylenebis(salicylaldehyde). Photos correspond to original methylenebis(salicylaldehyde):CEC, at a molar ratio of 1:50 (gelation time was 72 h), and hydrogel after dissolution in PBS buffers, with and without lysine addition, at a *w/v* ratio of 1:10 and 25 °C (**C**) [reprinted with permission from Copyright © 2021 by Svetlana Bratskaya et al. ref. [112] Licensee MDPI, Basel, Switzerland].

**Figure 3 gels-09-00277-f003:**
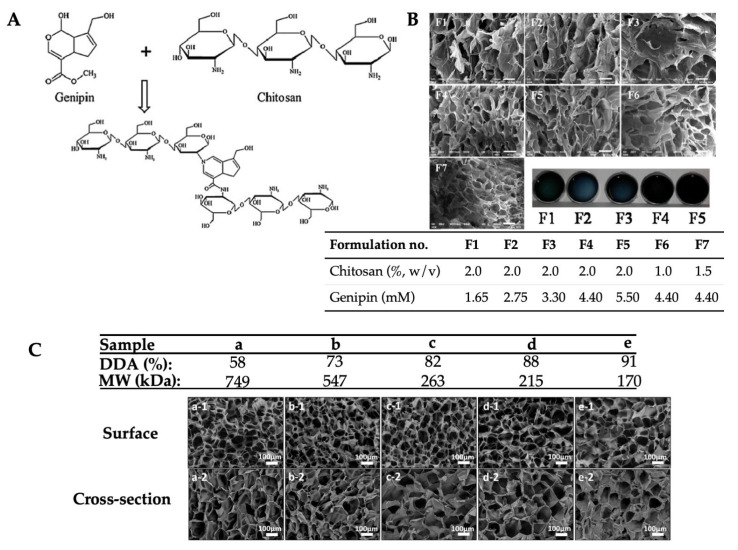
Mechanism of crosslinking between chitosan and genipin molecules (**A**), SEM images and photographs of genipin crosslinked chitosan hydrogels prepared by different formulations (**B**) (the scale bars represent 200 μm). The inserted table shows formulation codes (**F1**–**F7**) by using different concentrations of chitosan (85–95% DDA and MW of 300-450 kDa) and genipin [reprinted with permission from ref. [119] Copyright © 2014, Elsevier]. At 2% w/v chitosan and 0.02% w/v genipin (**C**), the different DDA and MW of chitosan are used to fabricate hydrogels, samples a–e (inserted table). SEM images show surface (a-1 to e-1) and cross-sectional (a-2 to e-2) morphologies of genipin crosslinked chitosan hydrogels (the scale bars represent 100 μm) [reprinted with permission from ref. [120] Copyright © 2021, Elsevier].

**Figure 4 gels-09-00277-f004:**
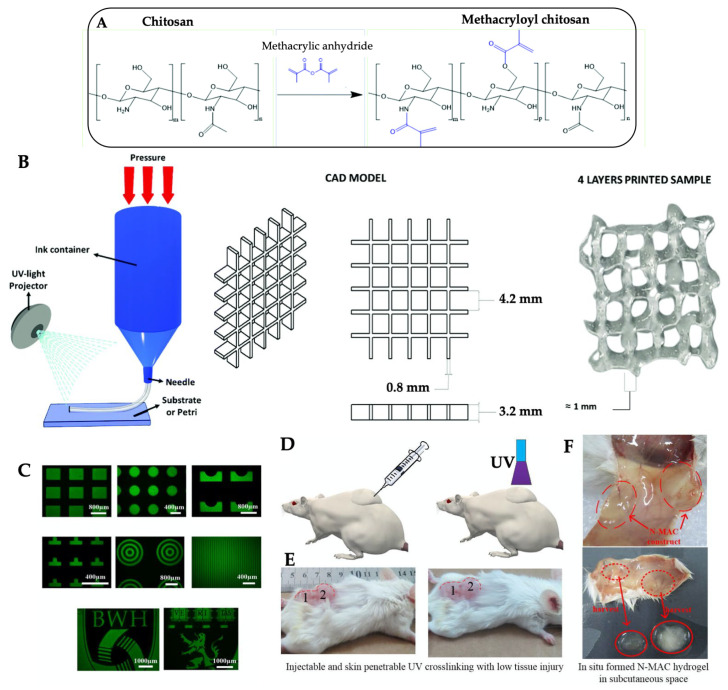
Scheme of methacrylation reaction of chitosan (**A**), extrusion-based 3D printing scheme (left), the used CAD model (middle), and the four-layer grid hydrogel immediately after printing (right) (**B**) [reprinted with permission from ref. [135] Copyright © 2022, the Royal Society of Chemistry]. Fluorescence images of chitosan hydrogel using photomask library (**C**), a schematic diagram of transdermal curing of injectable chitosan hydrogel, before (left) and after (right) skin-penetrable UV crosslinking (**D**), photos of mice (bottom, after shaving back feather) with injection of chitosan solution, before (left) and after (right) UV irradiation for 60 s (**E**), and typical photos of hydrogel adjacent with regional skin (up) and harvested hydrogel (down) after 6 days (**F**). [reprinted with permission from ref. [136] Copyright © 2015, Elsevier].

**Figure 5 gels-09-00277-f005:**
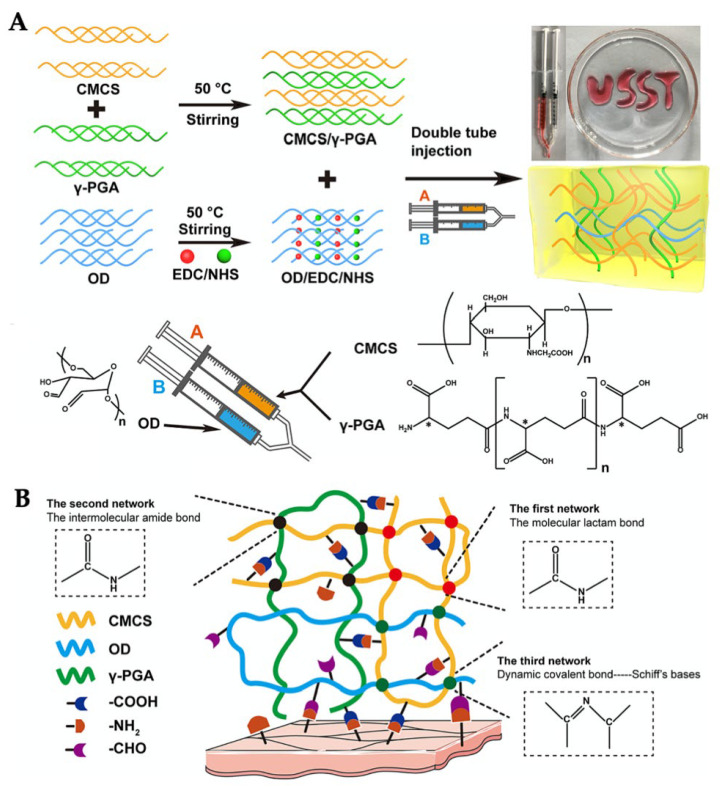
Schematic illustration of synthesis (**A**) and triple-network structure of carboxymethyl chitosans (CMCs), crosslinked by oxidized dextran (OD) and poly-γ-glutamic acid (γ-PGA) (**B**) (red dots: intramolecular amide bonds; black dots: intermolecular amide bonds; green dots: Schiff base bonds); [reprinted with permission from ref. [155,156] Copyright © 2022, Elsevier].

**Table 1 gels-09-00277-t001:** Dual crosslinking of hydrogels, prepared from chitosan and chitosan derivatives, and the resultant properties.

	First Crosslinking	Second Crosslinking	Properties	Ref.
Chitosan	sodium orthophosphate hydrate	genipin	8 min gelation time, 195 Pa storage modulus, ~100% swelling degree, and ~80% 3T3 cell viability	[144]
	2-formylphenylboronic acid	aldehyde	5 min gelation time, 648 Pa storage modulus, 110% swelling degree, and antifungal activity against *Candida* planktonic yeast	[145]
	acrylic acid	plasma treatment	5 min gelation time, effective surface functionalization	[146]
	disaccharide α,α′-trehalose derivative	citric acid	1 min gelation time, 100 Pa storage modulus.	[147]
	vanillin	microwave irradiation	2 min gelation time, 1.17 void fraction, 0.43 apparent density	[148]
	dihydrocaffeic acid	βGP	4 min gelation time, 500 Pa storage modulus, 30 kPa adhesive strength, 12% swelling degree, no cytotoxicity against 3T3 mouse fibroblasts, 100% lysozyme-containing simulated body fluid, hemostatic ability, and in vivo biocompatibility	[149]
Phenolic hydroxyl Chitosan	sodium tripolyphosphate	horseradish peroxidase	Fast gelation, 35% drug (5-FU) loading and 66.8% release	[150]
Carboxymethyl chitosan	citric acid	sodium alginate	~50% drug (carvacrol) loading and ~80% release, 20–100% swelling degree, 77% DPPH radical scavenging activity, antibacterial activities against *Escherichia coli* and *Staphylococcus aureus*, and ~80% HepG2 and HFL1 cell viability	[151]
	hyaluronic acid	EDC and NHS	~400 Pa storage modulus, 60% gel fraction, ~60% panax notoginseng saponins release, 90% L929 cell viability, anti-inflammatory property (30–40 μM iNOS secretion concentration), anti-oxidant property, and in vivo diabetic wound healing ability	[152]
	sodium alginate	sodium tripolyphosphate	6 min gelation time, 3–10 kPa storage modulus 100% MC3T3-E1 cell viability, in vivo biocompatibility, and in vivo bone healing	[153]
	sodium bicarbonate	tannic acid	100–200 Pa storage modulus, 80% DPPH radical scavenging activity, antibacterial activities against *Escherichia coli* and *Staphylococcus aureus*, 100% 3T3 cell viability, in vivo hemostatic ability, and in vivo wound healing ability	[154]
	oxidized dextran	poly-γ-glutamic acid	30 s gelation time, 2 Pa storage modulus, ~40% swelling degree, antibacterial activities against *Escherichia coli* and *Staphylococcus aureus*, 100% L929 cell viability, in vivo biodegradation, in vivo hemostatic ability, and in vivo wound healing ability	[155,156]

## Data Availability

Not applicable.

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
