# Peer review of "Recent Development of Functional Chitosan-Based Hydrogels for Pharmaceutical and Biomedical Applications"

_gels, 2023, doi:10.3390/gels9040277_

Round 1
Reviewer 1 Report
The manuscript entitled "Recent development of functional chitosan-based hydrogels for pharmaceutical and biomedical applications" is a very well documented review about the most important biomedical applications of chitosan hydrogels, consulting no less than 200 bibliographical references of recent date, demonstrating the authors' concern in this medical field.
The authors bring to our attention the importance of designing new localized drug delivery agents for the most effective medical therapy, of which smart hydrogels have a well-deserved place because they are able to respond to environmental stimuli or artificial triggers for drugs encapsulation/release. Biopolymeric hydrogels are preferred, and chitosan is one of the most used, after applying different chemical modifications designed to resolve its poor water solubility at neutral pH and grafting various agents in order to obtain the desired properties of chitosan depending on the pharmaceutical effect to be achieved.
The review is very well structured, mentioning a wide range of crosslinking agents along with the advantages and disadvantages of each technique, accompanied by very suggestive and easy to understand figures and tables. Each method aims for a short gelation time and a specific swelling degree, also being possible to combine two crosslinking techniques in order to improve mechanical properties, anti-inflammatory and antioxidant properties, in vivo diabetic wound healing ability, biocompatibility, antibacterial and antifungal activity, hemostatic ability, in vivo biodegradation, stability, rapid strain recoverability or self-healing ability.
The applicability in the medical field is very wide, chitosan-based hydrogels being mostly used as delivery systems, for a sustained or targeted release of the incorporated drug, as tissue engineering scaffolds, in disease treatments, and biosensors. The authors summarize some important examples of drugs that can be incorporated with chitosan, among them insulin, for which the major challenge is the oral administration.
The review is very rich in useful information and details, providing comprehensive, balanced and accurate information, is very well researched, but perhaps it would not be out of interest to re-evaluate the typesetting as there are a few small errors, e.g., a passage starting at line 456 is repeated from line 474.
Many drugs can improve their pharmacological and toxicological profile using these chitosan-based hydrogels, including oncological drugs, anti-diabetic drugs, anti-inflammatory drugs and many others. Hydrogels would therefore improve patient compliance and lead to successful treatment, but, as the authors conclude, there are still some challenges yet to be addressed in terms of stability, toxicity, biocompatibility and suitability for large scale production.
I consider the manuscript is well written, ready to be published and I congratulate the authors on their hard work.
Reviewer 2 Report
The article discussed the potential of chitosan-based hydrogels for use in pharmaceutical and biomedical applications. You described their use in drug delivery, tissue engineering, disease treatment, and biosensors. Although there are current challenges, these hydrogels have shown great potential for use in various medical applications.
But there are several other areas of use for chitosan-based hydrogels that are currently being explored. Here are a few additional examples:
Chitosan-based hydrogels have been investigated for their potential use in water treatment applications.
Chitosan-based hydrogels have been used to develop biosensors for various applications, including glucose monitoring for diabetes management and detection of biomarkers for disease diagnosis.
Chitosan-based hydrogels have been used in cosmetics as thickeners, stabilizers, and emulsifiers.
Chitosan-based hydrogels have also been investigated as tissue adhesives for surgical applications.
What do you think about this?
Here are a few things I would like to suggest. You just have to fix it.
line 34: involves in supramolecular ---> involve supramolecular
line 49: for a period of time ---> for some time
line 85: for development of hydrogels ---> for the development of hydrogels
line 91: orotic acid modified chitosan ---> orotic acid-modified chitosan
lines 456-473 and 474-495 were repeated.
Reviewer 3 Report
In this research, the authors reviewed the recent development of functional chitosan-based hydrogels for pharmaceutical and biomedical applications. Generally, it’s meaningful and interesting review. In my opinion, the current version of this manuscript fits the scope of the Gels and could be accepted after major revision.
My specific comments are in detail listed below:
1. Some minor mistakes existed in this paper. The authors should carefully check it, including (15–30 %w/w), Line 341.
2. In this review, some chitosan derivant and its gels was introduced. But, biguanide derived chitosan was not introduced. In my opinion, this part should be added. Some more relevant references should be added including 10.1016/j.carbpol.2022.119878, and 10.1016/j.ijbiomac.2020.07.029,.
3. Some chemical structures of the chitosan or chitosan derivants in the figures are of low quality. The authors should improve it carefully.
4. In the abstract (Line 43-52), the function and advantages of chitosan or chitosan derivants should more clearly revealed, especially its newly revealed PD-L1 depression capacity through AMPK activation. Some more relevant references should be added 10.1016/j.cej.2022.140164, 10.1016/j.molcel.2021.03.037, and 10.1016/j.immuni.2016.02.004.
5. Better future perspective should be added, especially the prospects of chitosan based hydrogels for clinical transformation.
Round 2
Reviewer 3 Report
The current version of this paper could be accepted.